# A Novel Aerodynamic Modeling Method Based on Data for Tiltrotor evtol

**Haiyang Wang** [1] , **Peng Li** [1,*] and **Dongsu Wu** [2]

1   College of Automobile and Traffic Engineering, Nanjing Forestry University, Nanjing 210037, China; wanghaiyang1@njfu.edu.cn
2   College of Civil Aviation, Nanjing University of Aeronautics and Astronautics, Nanjing 210016, China; tissle@nuaa.edu.cn
*   Correspondence: dawan007@njfu.edu.cn

**Abstract:** A data-driven aerodynamic modeling method is proposed to address the problem that traditional modeling methods based on physical mechanisms cannot fully represent the special aerodynamic characteristics of tiltrotor evtol aircraft. By analyzing the uniquely complex aerodynamic characteristics of electric vertical take-off and landing (evtol) aircraft, an MLP neural network model has been constructed that reflects the coupling characteristics between influencing factors. Using the XV15 wind tunnel test data, a dataset was constructed, and the neural network model was trained and validated. Simulation results show that the selected data-driven method can accurately predict the aerodynamic characteristics of the longitudinal transition phase of the tiltrotor evtol.

**Keywords:** aerodynamic modeling; data-driven; neural networks; tiltrotor; transition phase





## 1. Introduction

Urban traffic congestion has worsened due to increasing urbanization. This has led to the development of electric vertical take-off and landing (evtol) aircraft concepts [1,2]. The tiltrotor evtol configuration, which combines features from traditional fixed-wing and rotor aircraft, shows potential for sustainable transport over medium to long distances, particularly in areas with complex routes and severe congestion [3].

Aerodynamic modeling is critical in determining the dynamic characteristics and simulation fidelity of maneuvering stability for evtol. Furthermore, aerodynamic modeling can offer an affordable and replicable simulation platform for designing flight control systems, assessing maneuvering stability, modifying designs, and other relevant aspects of evtol operations. Concurrently, aerodynamic modeling is the primary method used to describe evtol aerodynamic effects [4]. It reflects the variation characteristics of these effects with factors such as airspeed, configuration, tilt angle, and control surface deflection angle.

The aerodynamic characteristics of tiltrotor evtol aircraft vary significantly between different flight modes, particularly during the transition phase caused by nacelle tilting [5,6]. Due to the influence of flight conditions and rotor tilting motions, tiltrotor evtol aircraft exhibit high complexity, strong coupling, and nonlinearity in their aerodynamic characteristics. These variations in rotor-induced velocities, as well as the effects of rotor downwash and wake, make aerodynamic modeling extremely challenging.

Some aerodynamic propulsion modeling methods for lift + cruise evtol aircraft are proposed in references [7–10]. System identification techniques and computational fluid dynamics (CFD) simulations are used to develop flight dynamic models and validate the predictive capabilities of aerodynamic propulsion models. However, aerodynamic modeling methods for tiltrotor evtol still rely on traditional physical mechanisms for aerodynamic modeling. References [11,12] suggest that traditional methods are not effective in simulating certain aerodynamic characteristics, such as rotor wake and downwash. Concurrently, there is a certain contradiction between the realism of models established by traditional

methods and the real-time performance of simulations. Therefore, most aerodynamic modeling methods used in real-time simulation environments are based on simplified physical mechanisms, which limits their ability to perform high-fidelity simulations.

The use of algorithms and computational capabilities has led to the widespread adoption of data-driven aerodynamic modeling methods based on neural networks, commonly used neural network methods in aerodynamic prediction include MLP, RBFNNs, CNNs, RNNs, and GANs [13–15]. These methods do not require the establishment of complex mathematical formulae based on physical mechanisms [16,17]. Instead, they learn the hidden dynamic characteristics of the system through sample data. Data-driven methods can address high-dimensional, multiscale, and nonlinear problems that are difficult to solve with traditional methods.

The aim of this study is to validate the use of data-driven modeling methods in aerodynamic modeling during the transition phase of tiltrotor evtol aircraft. This study analyses the complex relationship between dimensionless aerodynamic force coefficients of tiltrotor aircraft and their influencing factors. A multilayer perceptron (MLP) neural network is selected for multivariate nonlinear regression for data-driven aerodynamic modeling. The data-driven model was constructed from the wind tunnel test data of XV-15 and trained using an MLP neural network. A comparison was conducted between the predictive performance of a data-driven model and a mathematical model based on physical mechanisms. The results demonstrate that the data-driven model can capture aerodynamic characteristics that are challenging to express in mathematical models.

This paper is structured as follows: Section 2 introduces the relationship between dimensionless aerodynamic coefficients of tiltrotor evtol and their influencing factors. Section 3 describes the advantages of data-driven methods based on MLP neural networks and the construction of the dataset. Section 4 presents the evaluation criteria and training methods for the neural network. The predictive performance of the data-driven method is compared with that of mathematical models to validate the accuracy of the established neural network model. Conclusions are drawn in Section 5.

## 2. Analysis of Aerodynamic Characteristics of Tiltrotor evtol

Tiltrotor evtol aircraft have unique aerodynamic characteristics, including interactions between numerous control surfaces, rotor–fuselage interactions, rotor–rotor interactions, and rapid changes in aerodynamics during the transition phase as the rotor pitch angle varies. Therefore, a suitable aerodynamic modeling strategy for tiltrotor evtol must combine aspects of traditional aircraft modeling strategies. Figure 1 shows the tiltrotor evtol Vahana that Airbus conducted research and testing on several years ago.

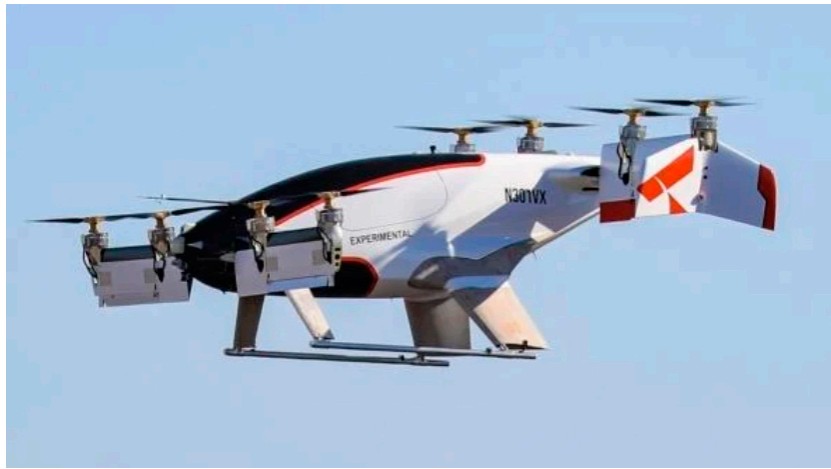

**Figure 1.** Test evtol for Airbus A3: Vahana.

Aerodynamic modeling for fixed-wing aircraft usually entails generating aerodynamic interpolation tables or representing dimensionless aerodynamic force and moment coefficients as functions of aircraft states and controls [8]. The aerodynamic force and moment coefficients, which are dimensionless, are commonly expressed as functions of various parameters, including angle of attack ($\alpha$), sideslip angle ($\beta$), angular rate ($p, q, r$), and control surface deflections (elevator deflection angle $\delta_e$, aileron deflection angle $\delta_a$, rudder deflection angle $\delta_r$). These coefficients are known as response variables, while explanatory variables include factors such as airflow angle. The dimensionless aerodynamic forces and moment coefficients of a fixed-wing aircraft, such as lift coefficient $C_L$, drag coefficient $C_D$, and pitch moment coefficient $C_M$, can be expressed as follows:

$$C_L, C_D, C_M = f(\alpha_F, \beta_F, U, V, W, p, q, r, \ \delta_e, \delta_a, \delta_r \ldots)$$

Rotorcraft aerodynamic modeling relies heavily on computational or flight test data due to the difficulty in scaling rotorcraft proportionally and equipment limitations for wind tunnel testing [8]. It is important to note that unlike fixed-wing aircraft, rotorcraft aerodynamic modeling requires a different approach due to their unique characteristics [18]. Since stability axes and wind axes become undefined in hover, modeling is generally only performed in the body axes for rotorcraft. The formulation in terms of body-axis velocity components, as opposed to airflow angles $\alpha$ and $\beta$, allows the state variables to be defined in hover and reflects the fact that fuselage angle of attack and angle of sideslip are less physically meaningful for describing rotorcraft aerodynamics [8]. Explanatory variables for rotorcraft modeling often include body-axis velocity components ($U, V, W$), angular rates ($p, q, r$), pilot control inputs, and rotor states such as flapping and inflow.

Tiltrotor evtols currently have smaller rotor diameters and do not use cyclic pitch control, which makes rotor section modeling less critical, except for specific rotor states [19]. Therefore, the aerodynamic modeling of tiltrotor evtol includes most of the characteristics of fixed-wing aircraft. However, the aerodynamic characteristics of the rotor section are more evident in the fuselage–rotor interaction. Figure 2 shows the relationship between the dimensionless aerodynamic coefficients and moments of tiltrotor evtol and the control and state variables.

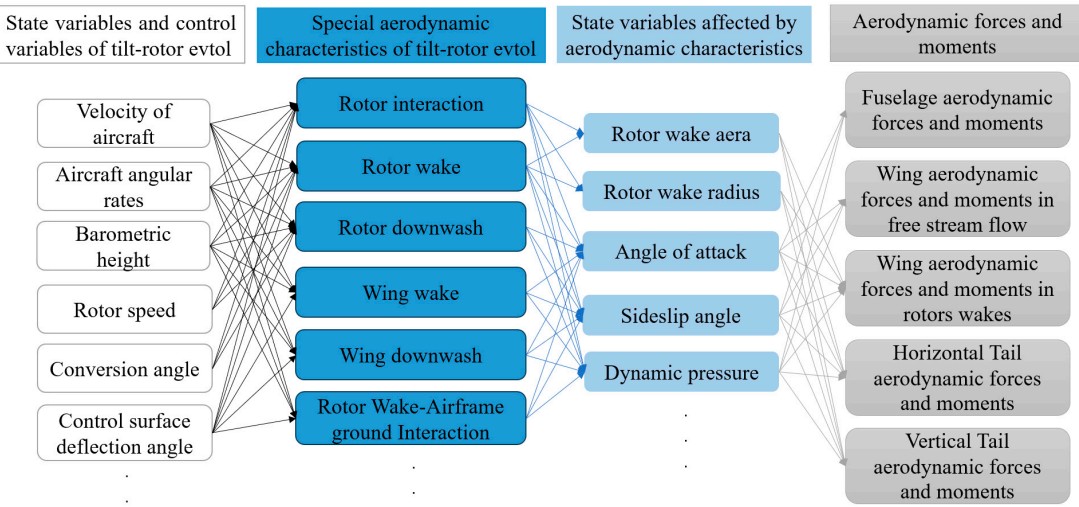

**Figure 2.** Coupling relationship of aerodynamic characteristics of tiltrotor evtol.

Variations in the state and control variables of tiltrotor evtol can cause changes in airflow speed and direction, as well as induced velocity characteristics, across different parts of the fuselage. These changes can result in complex aerodynamic features, such as rotor wake and rotor–fuselage interactions, which ultimately affect the aerodynamic forces of different components. It is important to note that these aerodynamic characteristics are

interdependent, which means characteristics located in the second layer of Figure 1 can impact each other.

However, most aerodynamic modeling work for evtol to date has used analytical or semiempirical models for research or application purposes. These traditional methods cannot fully express certain special aerodynamic characteristics, such as the impact of rotor wakes on different parts of the fuselage, which greatly simplifies highly complex aerodynamics. The discrepancy between mathematical models and wind tunnel data is evident in the limitation of accuracy. Figure 3 shows this for the gravity-to-lift ratio of the XV15 tiltrotor aircraft versus airspeed [20], highlighting the inability of mathematical models to fully capture the effects of rotor wakes.

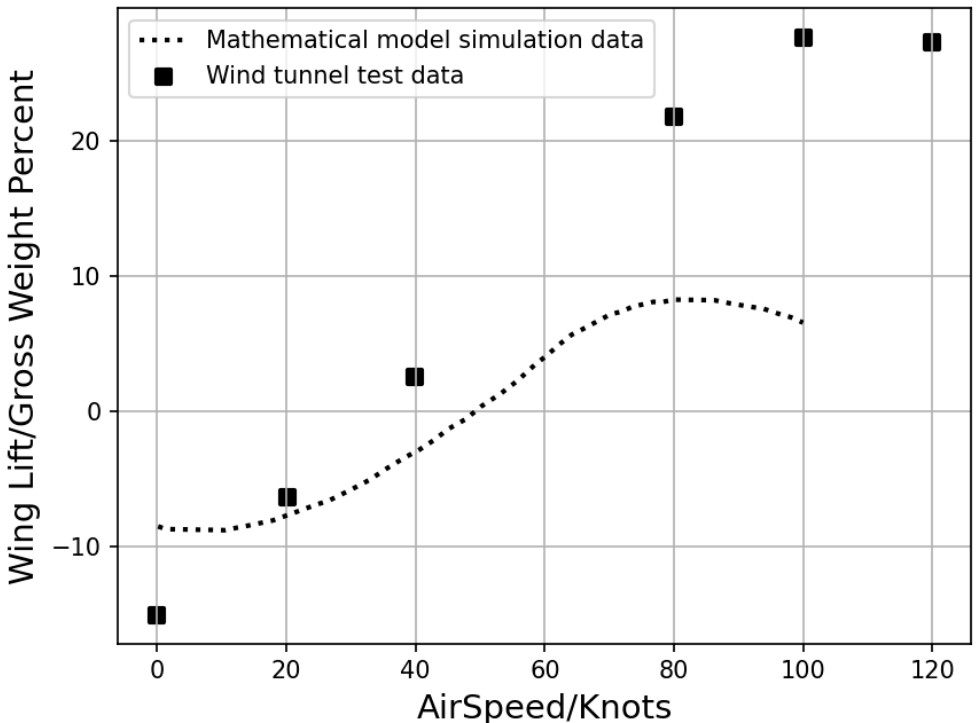

**Figure 3.** Effect of rotor wake on wing lift of XV15 tiltrotor aircraft.

## 3. Data-Driven Modeling Methods

The analysis of the mechanism and influencing factors of the transition phase of the tiltrotor evtol in the Section 2 reveals that its complex aerodynamic characteristics result from the interaction of multidimensional factors. This interaction determines a nonlinear relationship between the input and output of the aerodynamic characteristic model that needs to be established. This type of relationship is reflected not only between factors but also between layers. Therefore, we selected an MLP neural network for aerodynamic modeling. This type of network is capable of solving complex nonlinear relationships.

### 3.1. Data-Driven Network Model Design

The multilayer perceptron (MLP) is a type of feedforward neural network with a simple connectivity pattern. It comprises an input layer, one or more hidden layers, and an output layer. Each hidden layer contains multiple neurons [21]. The MLP operates by connecting each neuron in a layer to every neuron in the next layer with appropriate weights and biases. Input data propagate from the input layer to the output layer through forward propagation.

The input data are received by the network through the input layer neurons. Each neuron in a hidden layer receives a weighted sum of signals from all neurons in the previous layer. This weighted sum is then passed through an activation function, such as sigmoid or ReLU, which introduces nonlinearity and enables the network to learn complex patterns. The output of the activation function becomes the output of that neuron and the input for neurons in the next layer. MLP calculates its error by comparing its predictions with the expected outputs using a loss function at the output layer. The error is then propagated backward through the network using backpropagation, which employs the chain rule to calculate the contribution of each weight to the error. The weights in each layer are then adjusted based on the error signal and a learning rate to minimize the overall error. The iterative process of forward pass, error calculation, and backpropagation with weight updates continues during training until the network achieves an acceptable level of performance [22].

MLP networks can be used for multifunctional learning, including classification and regression tasks. MLP demonstrates proficient performance in handling multivariable nonlinear regression problems with its classic architecture [21]. The relationship between the dimensionless aerodynamic coefficients introduced in Section 2 is similar in structure to the MLP neural network and essentially belongs to multiple nonlinear regression. Therefore, this article introduces an MLP neural network based on the input–output and influence factors described in the Section 2. The hidden layer neurons represent the unique aerodynamic characteristics and state variables of the tiltrotor evtol. However, in these complex coupling relationships, some factors do not influence each other. Therefore, these unrelated neurons should not be connected. The MLP topology diagram for the unique aerodynamic characteristics of the tiltrotor evtol is obtained as Figure 4.

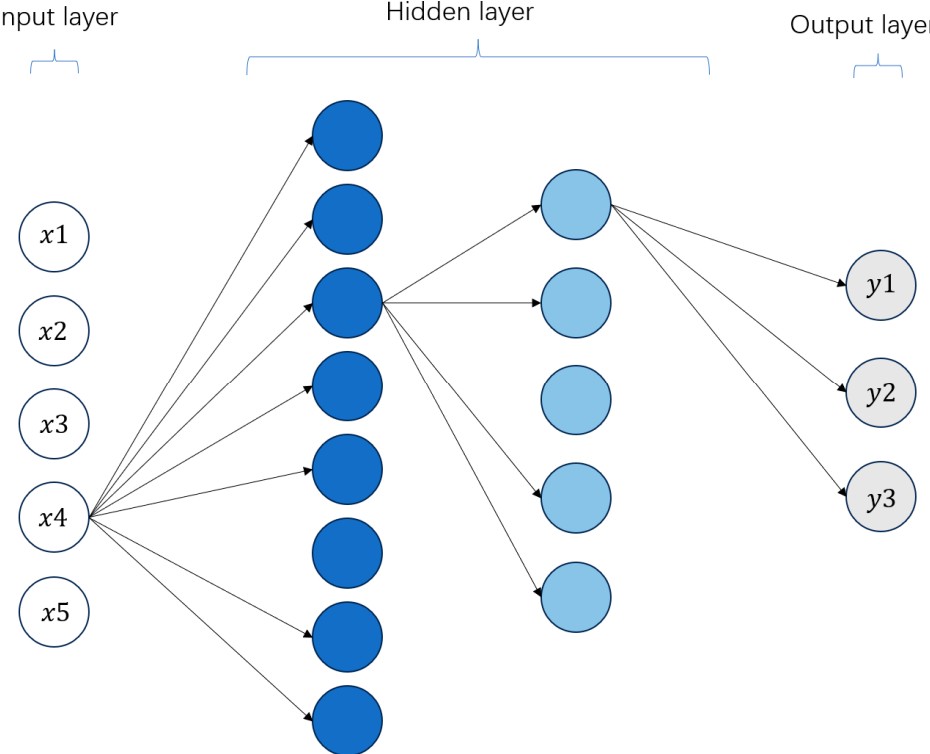

**Figure 4.** The structure of the MLP network topology (connections between neurons are only partially displayed for reasons of clarity and comprehensibility).

During network training, the structural parameters of the MLP are determined based on the coupling characteristics of aerodynamic features and their training effect. To achieve non-interconnected neurons, the weight of the corresponding connections in the weight matrix can be set to zero. This reflects the unique aerodynamic characteristics of the tiltrotor evtol.

This article employs a network comprising multiple layers of neurons, each comprising a LeakyReLU activation function. The network is trained using the Adam optimizer to calculate the weights of the MLP neural network model. All of these components are implemented in Python code based on the PyTorch framework.

*3.2. Sample Data Requirements and Generation*

The aerodynamic characteristics and rotor layout of the XV15 tiltrotor aircraft are similar to some tiltrotor evtols, resulting in largely consistent aerodynamic characteristics. This article conducts data-driven aerodynamic modeling on the tilting rotor evtol, based on wind tunnel test data of the transition maneuver of XV15. Due to space limitations, this article focuses on data-driven model construction for longitudinal aerodynamics, the generation of sample data, model training, and validation.

According to reference [20,23], for the longitudinal transition maneuver of the XV15 tiltrotor aircraft, aerodynamic forces and moments ($C_L$ $C_D$ $C_M$) are defined as functions of a set of flight state values ($M_N$ $\alpha_F$ $\beta_M$ $F_X$ $\delta_e$). Among them, $M_N$ is the Mach number, $\alpha_F$ is the angle of attack of the fuselage, $\beta_M$ is the tilt angle of the nacelle, $F_X$ is the flap/aileron angle mode selection, and $\delta_e$ is the elevator deflection angle. During the transition phase, the value ranges of variables mentioned above for XV15 are as Table 1.

**Table 1.** Range of explanatory variables for the transition phase of XV15 tiltrotor aircraft.

| Input Variables | Minimum | Maximum |
|---|---|---|
| $M_N$ | 0 m/s | 120 m/s |
| $\delta_e$ | $-20°$ | $20°$ |
| $F_X$ | $F_1$ ($0°/0°$) | $F_2$ ($40°/25°$) |
| $\beta_M$ | $0°$ | $90°$ |
| $\alpha_F$ | $-15°$ | $15°$ |

By considering the range constraints of explanatory variables during the transition phase of the tiltrotor evtol and the inclined transition corridor determined in reference [20,23], a multidimensional sample space can be established. This sample space allows for a one-to-one correspondence between the aerodynamic coefficients and the explanatory variables. Additionally, there are constraints between variables, in addition to the range constraints of the variables themselves. Figure 5 shows the transition corridor resulting from the constraint between the tilt angle $\beta_M$ and the horizontal velocity $V_X$. When the nacelle inclination angle is $0°$, XV15 is in helicopter mode, and when the nacelle inclination angle is $90°$, it is in aircraft mode. The dataset needed for the neural network is constructed by selecting points from a vector space consisting of flight envelopes and constraints in multiple dimensions. It is important to note that the aerodynamic coefficient profile has both linear and nonlinear regions that correspond to a single explanatory variable. The nonlinear region has a higher density of points compared with the sparser points in the linear region. The constructed sample dataset comprises 2000 sample points, with each point represented in the form of $C_L, C_D, C_M = f(M_N \ \alpha_F \ \beta_M \ F_X \ \delta_e)$.

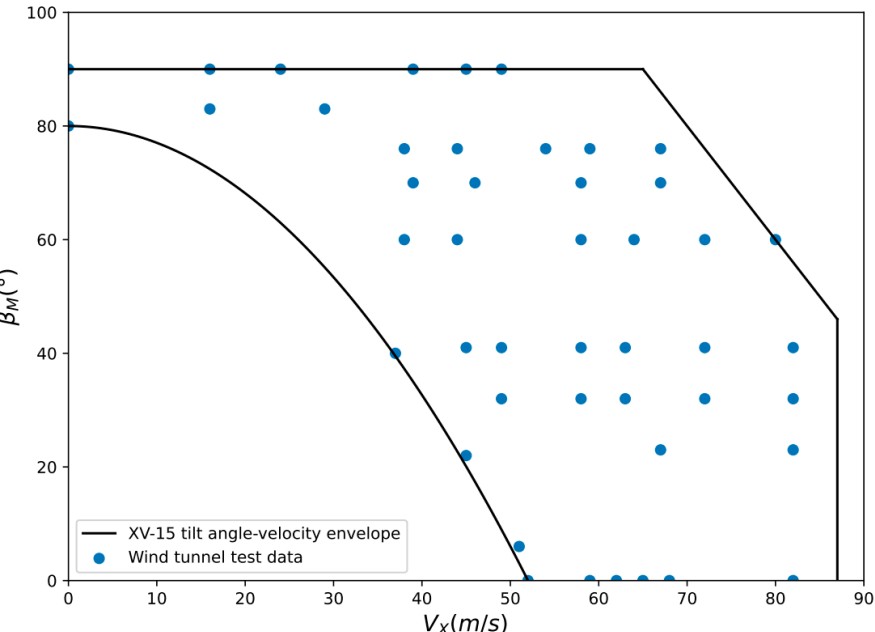

**Figure 5.** XV-15 tilt angle–velocity flight envelope.

## 4. Results and Discussion

This section trains and validates the aerodynamic model using an MLP neural network based on the sample dataset obtained from the XV15 transition maneuvers in Section 3. As the data in this article are only a small-scale sample dataset, they were divided into training, validation, and testing sets using the commonly used 7:2:1 ratio. The training set is used as input for model training, while the validation set is used to select appropriate model hyperparameters. The test set is utilized to validate the predictive performance of the trained model without being involved in model construction [24].

### 4.1. Network Training Method and Results

The process of constructing a data-driven model involves a crucial step in the selection of appropriate hyperparameters, which necessitates the adoption of appropriate criteria for evaluating the predictive efficacy of different hyperparameter combinations. In this study, relative error and coefficient of determination are used as performance metrics to evaluate the network. The relative error is defined as

$$error = \frac{||y_{pre} - y_t||_F}{||y_t||_F} * 100\%$$

where $y_{pre}$ is the predicted output of the network, and $y_t$ is the true value, yielding an aggregate relative error over the entire performance of the network. Specifically, for the lift coefficient ($C_L$), the individual errors for each aerodynamic coefficient are described as follows:

$$error_L = \frac{\left||C_{L_{pre}} - C_{L_t}\right||_2}{||C_{L_t}||_2} * 100\%$$

Meanwhile, the coefficient of determination is defined as

$$R^2 = 1 - \frac{SS_T}{SS_R}$$

$$SS_T = \sum_{t=1}^{k}\left(C_{L_t} - C_{L_{pre}}\right)^2$$

$$SS_R = \sum_{t=1}^{k} \left( C_{L_t} - \overline{C_{L_t}} \right)^2$$

where $C_{L_t}$ is the true lift coefficient, and $C_{L_{pre}}$ is the network predicted lift coefficient. The relative error can indicate how well a trained neural network behaves on new data that were not used in training. It can be thought of as a curve fitting, where the relative error represents the average deviation from the fitted curve to the wind tunnel test data. The coefficient of determination is used to assess how well a regression model explains the relationship between a dependent variable and an independent variable. It reflects how well the model fits the data it was trained on. An acceptable relative error of 5% [11] or less is based on established evtol aerodynamic models. A coefficient of determination closer to 1 indicates a stronger relationship between the variables, meaning the model explains most of the variation in the dependent variable.

For an MLP neural network, key hyperparameters that influence prediction performance include the number of hidden layers, neurons per layer, activation functions, loss function types, training epochs, and batch size. This article utilizes the random search method in Python to randomly sample hyperparameter values from a defined range and compare the above evaluation criteria to select the optimal hyperparameter. Note that this article uses the average of the errors and determination coefficients for each aerodynamic coefficient as the evaluation criterion, due to their varying relative errors and determination coefficients under different hyperparameters. The results of training the MLP neural network with varying hyperparameter values for predicting lift coefficient errors are tabulated below as Table 2.

**Table 2.** Different combinations of MLP training parameters.

| Hyperparameters | MLP | | | | | | | | |
|---|---|---|---|---|---|---|---|---|---|
| Number of Epochs | 100 | 100 | 100 | 50 | 150 | 100 | 100 | 100 | 100 |
| Batch Size | 10 | 20 | 30 | 20 | 20 | 20 | 20 | 20 | 20 |
| Number of Neurons/hidden Layers | 64/2 | 64/2 | 64/2 | 64/2 | 64/2 | 32/2 | 96/2 | 64/1 | 64/3 |
| Relative error | 1.625% | 1.361% | 1.455% | 1.413% | 1.327% | 1.314% | 1.391% | 1.861% | 1.937% |
| Coefficient of Determination | 0.9921 | 0.9967 | 0.9925 | 0.9981 | 0.9986 | 0.9972 | 0.9975 | 0.9907 | 0.9943 |

In summary, by iteratively simulating the neural network and evaluating the relative error and coefficient of determination across different network hyperparameter settings, selecting the hyperparameter combination with the minimum relative error and the maximum coefficient of determination, the optimal values for the number of layers, neurons, and other hyperparameters for the MLP neural network are derived, as shown in the accompanying Table 3.

**Table 3.** Optimal network parameter combination.

| Parameters | MLP |
|---|---|
| Number of inputs/outputs | 5/3 |
| Number of Epochs | 100 |
| Batch Size | 20 |
| Number of Layers/Neurons | 32/2 |
| Sequence Length | - |
| Activation Functions | Leakyrelu |
| Loss Function | L1Loss |

**Table 3.** *Cont.*

| Parameters | MLP |
| --- | --- |
| Relative error | 1.314% |
| Coefficient of Determination | 0.9972 |
| Total training time | 395.43 s |

### 4.2. Network Effect Verification

Based on the analysis of dimensionless aerodynamic coefficients and their influencing factors in Section 2, it can be concluded that the rotor wake is the most significant aerodynamic characteristic affecting the transition phase of the XV15. This, in turn, affects the aerodynamic forces on other parts of the fuselage. The mathematical model report and wind tunnel test data of the XV15 tiltrotor aircraft support this conclusion. The tilt angle is the explanatory variable that has the greatest impact on the rotor wake. Therefore, wind tunnel test data under different tilt angle conditions were selected as the validation set to verify the network's predictive effect. The validation results compare wind tunnel data, results from the mathematical model established in the literature [9], and predictions from neural network models. The purpose is to illustrate the difference between data-driven modeling methods and models established using traditional physical formulas.

Figures 6–8 present the predicted results for the selected transition conditions. Comparisons are made between traditional models and neural networks under different flap/aileron angle mode selection, Mach numbers, and elevator deflection angles. Figure 6 shows that the flap/aileron angle mode selection has a significant impact on the aerodynamic coefficient during the transition phase under low Mach number conditions. Increasing the angle of the flaps/ailerons to 40/25° results in a nearly 50% increase in the aerodynamic coefficient compared with the original. Figure 7 demonstrates that the impact of the Mach number on the aerodynamic coefficient is relatively small when it is low. The available wind tunnel test data only include the relationship between Mach number and aerodynamic coefficient. Furthermore, the tiltrotor evtol typically operates at low speeds or Mach numbers. Therefore, this article compares the aforementioned relationships. Figure 8 illustrates the impact of the elevator deflection angle on the lift coefficient and aircraft stall angle of attack. The results indicate that a 20° increase in elevator deflection angle leads to a 40% decrease in lift coefficient and a corresponding increase in the aircraft stall angle of attack.

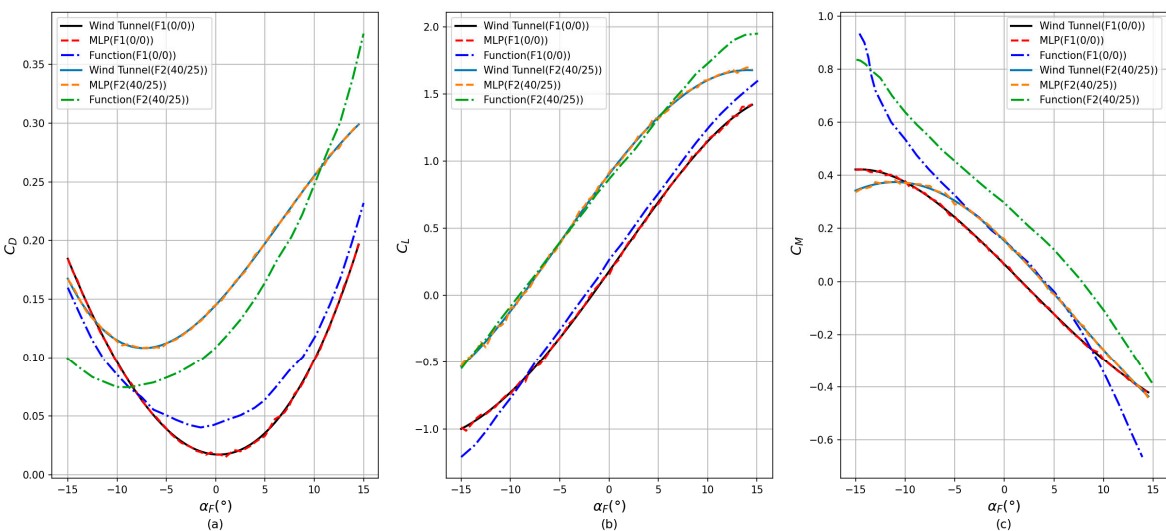

**Figure 6.** Relationship between angle of attack and selected flap/aileron angle with aerodynamic coefficient at a tilt angle of 0°. (**a**) Drag coefficient. (**b**) Lift coefficient. (**c**) Pitch moment coefficient.

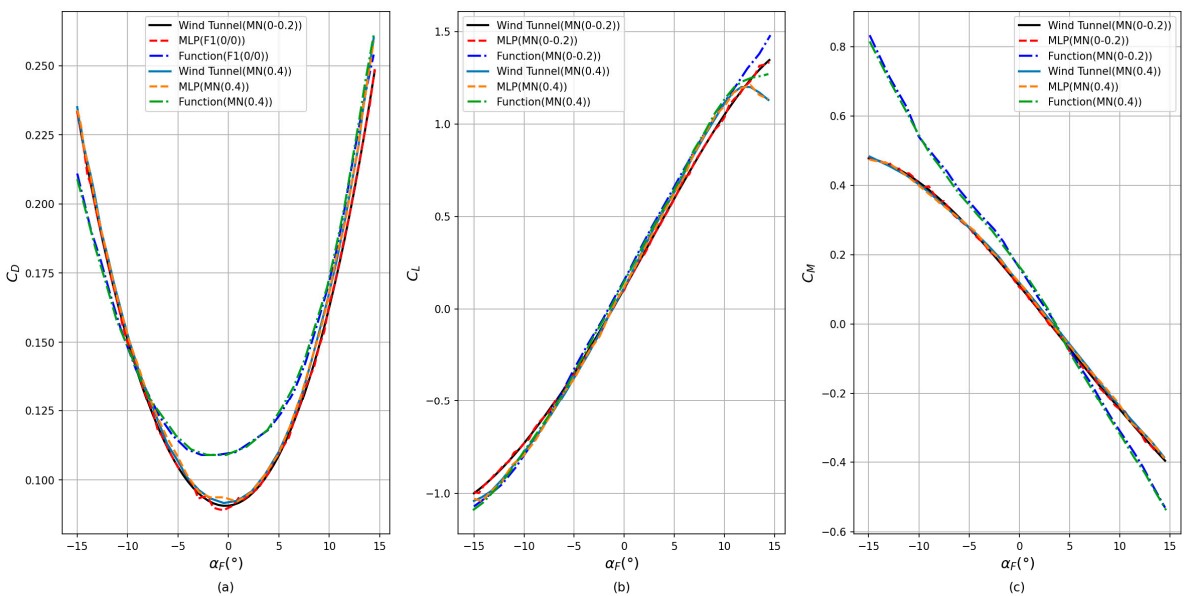

**Figure 7.** Relationship between angle of attack and Mach number with aerodynamic coefficient at a tilt angle of $30°$. (**a**) Drag coefficient. (**b**) Lift coefficient. (**c**) Pitch moment coefficient.

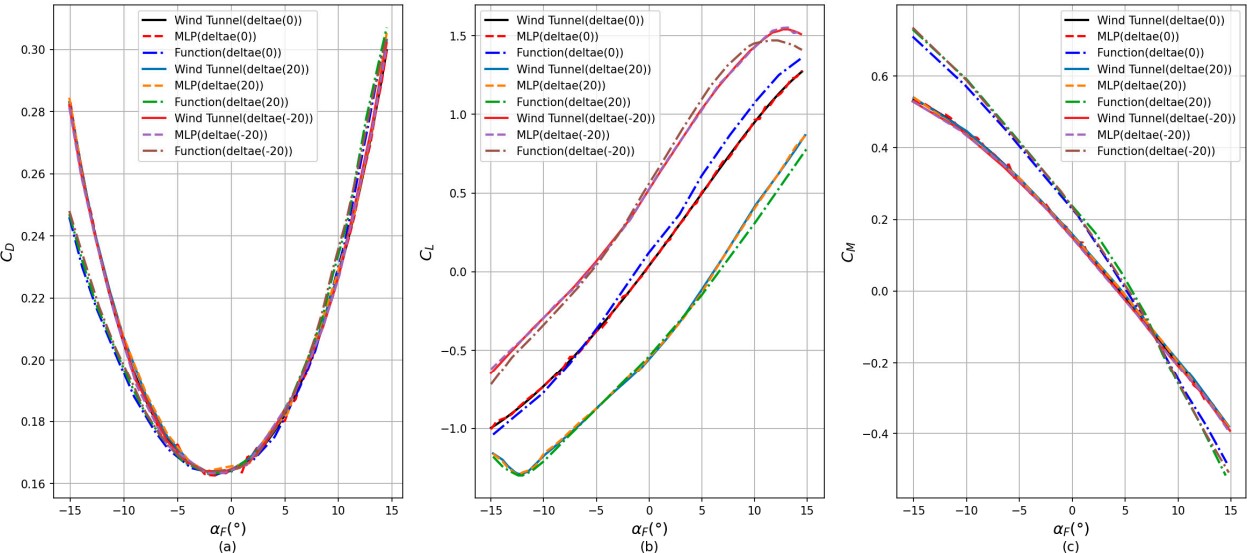

**Figure 8.** Relationship between angle of attack and angle of elevator deflection with aerodynamic coefficient at a tilt angle of $60°$. (**a**) Drag coefficient. (**b**) Lift coefficient. (**c**) Pitch moment coefficient.

It is evident that there is a significant discrepancy between the predicted data from the mathematical model and the wind tunnel data. The data-driven approach more accurately reflects the nonlinear aerodynamic characteristics during the transition phase, as indicated in the literature [19], which states that the mathematical model does not provide an accurate description of these rotor wake and stall phenomena.

Table 4 displays the coefficient of determination $R^2$ and the relative error of the predicted data from the MLP neural network. The results indicate that the aerodynamic coefficient predicted by the network closely matches the wind tunnel data, with an error of approximately 2%. Meanwhile, the coefficient of determination for network prediction is close to 1, indicating that the model explains most of the variation in the dependent variable. In contrast, the prediction of aerodynamic coefficients based on mathematical models has an error of about 10%, and the accuracy of determining coefficients is relatively low. This level of accuracy meets the requirements for aerodynamic models reported in

the literature [11]. The results demonstrate that, in general, data-driven methods based on fully connected neural networks provide better prediction results than models built using traditional methods.

**Table 4.** Comparison of MLP and Mathematical Model Errors.

| Test Set | Coefficient of Determination (MLP) $C_L$ $C_D$ $C_M$ | Relative Error (MLP) $C_L$ $C_D$ $C_M$ | Coefficient of Determination (Mathematical Model) $C_L$ $C_D$ $C_M$ | Relative Error (Mathematical Model) $C_L$ $C_D$ $C_M$ |
|---|---|---|---|---|
| 1 | 0.9971/0.9962/0.9987 | 1.474%/1.032%/0.861% | 0.9258/0.9463/0.9731 | 8.326%/7.034%/15.338% |
| 2 | 0.9916/0.9965/0.9984 | 1.177%/0.946%/1.052% | 0.9341/0.9480/0.9712 | 4.584%/4.872%/9.635% |
| 3 | 0.9959/0.9963/0.9989 | 0.986%/0.912%/0.883% | 0.9246/0.9384/0.9658 | 4.669%/8.453%/11.763% |

## 5. Conclusions

A neural network structure that conforms to the aerodynamic characteristics and mechanism of the tiltrotor evtol was constructed using an MLP neural network. To create a dataset for neural network training and validation, wind tunnel test data from the XV15 tiltrotor with similar aerodynamic characteristics and structure were selected. The dataset was constructed by analyzing the sample data structure and constructing a sample vector space with constraints. Finally, the optimal hyperparameter combination for the neural network was selected by designing evaluation indicators. Predictions were then made based on the test set under different tilt angle conditions. The simulation results show the following:

1. Neural-network-based data-driven methods can accurately predict the aerodynamic characteristics of tiltrotor evtol aircraft during the transition phase, with an error rate of less than 2% compared with wind tunnel test data.
2. Traditional physical modeling methods exhibit larger errors in representing the complex aerodynamic characteristics of tiltrotor aircraft due to the limitations of mathematical formulae, typically up to nearly 10%.
3. The MLP neural network structure constructed based on the aerodynamic characteristic mechanism of the tiltrotor evtol is effective.

In summary, this study shows that data-driven methods using neural networks can predict the aerodynamic characteristics of tiltrotor evtol aircraft during the transition phase. Due to the high fidelity fitting and relatively simple modeling process of MLP for nonlinear and strongly coupled aerodynamic characteristics, this method can be extended and applied to the full degree of freedom and full flight envelope aerodynamic modeling of tiltrotor evtol aircraft.

**Author Contributions:** Writing—original draft preparation, H.W.; writing—review and editing, P.L. and D.W. All authors have read and agreed to the published version of the manuscript.

**Funding:** This research was funded by the China Scholarship Council, grant number 202308320187.

**Data Availability Statement:** The data presented in this study are available in [NASA] at [https://ntrs.nasa.gov/citations/19730022217], reference number [20].

**Conflicts of Interest:** The authors declare no conflict of interest.

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
