# Peer review of "A Novel Aerodynamic Modeling Method Based on Data for Tiltrotor evtol"

_applsci, doi:10.3390/app14104055_

Round 1
Reviewer 1 Report
Comments and Suggestions for Authors
Dear Authors
I have read your paper proposal with interest. I believe the subject is interesting and will most likely grow substantially in the future. I believe the story and the issues of the V-22 Osprey should be mentioned. It is much bigger and non-electrical but it was handled by a much bigger team with much bigger budget and some of the problems you faced were never fully solved. Add some references connected to V-22.
Figure 2 left or lift?
Table 1 minimum or maximum
And the biggest issue. Can you give some information about the size of the wind tunnel and your model?
Is there any special reason why you did comparison on Mach number. Usually it is important when you are close to Mach 1 (or al least over 0.5)
Best regards
Author Response
Dear reviewer: Thank you for your decision and constructive comments on my manuscript. We have carefully considered the suggestion of Reviewer and make some changes. We have tried our best to improve and made some changes in the manuscript.Our response and modifications to the questions raised are as follows: 1.The word on the vertical axis in Figure 2 should be 'lift' 2.The third column of Table 1 should be the maximum value 3.For the experimental data and some formulas in the article, they are mainly from the references: Harendra P B, Joglekar M J, Gaffey T M, et al. V/STOL tilt rotor study. Volume 5: A mathematical model for real time flight simulation of the Bell model 301 tilt rotor research aircraft[R]. 1973. The wind tunnel models include: Ames 40- by 80-Foot Wind Tunnel 4.The reason why we chose to compare Mach numbers is that there is almost no data directly related to airspeed and aerodynamic coefficients in the corresponding experimental data available. Therefore, we chose to analyze the low Mach number situation, which precisely corresponds to the actual basic operating speed of evtol.We have included these descriptions in the analysis of the corresponding experimental results(286-289, p9) Best regards

Reviewer 2 Report
Comments and Suggestions for Authors
This paper is interesting for readers, researchers, and companies. It discusses about a neural network structure that conforms to the aerodynamic characteristics and mechanism of the tiltrotor evtol was constructed using an MLP neural network.
Some comments below:
Add pictures to descrie the tiltrotor evtol
The quality of figure 2 is poor
Table : maybe the second decimal is enough
Add a list of symbols
Figure 5, 6 and 7 are interesting and they need more space in the manuscript: please use one panel for each graph and please provide a careful discussion for each new figure. Authors can save the space by deleting Table 3 and 4 (i.e. add information in the text).
References about ANN can be improved by discussing important papers in the field of civil engineering to give a complete overview for its application. Maybe it is useful to discuss results given by the quite recent paper below:
F. Rizzo, L. Caracoglia, 2021. Examination of Artificial Neural Networks to predict wind-induced displacements of cable net roofs, Engineering Structures, 245, 112956
Please add information about WT tests, add pictures, and models description and more and more details.
Please provide information about Matlab subroutines used to the ANN.
Author Response
Dear reviewer: Thank you for your decision and constructive comments on my manuscript. We have carefully considered the suggestion of Reviewer and make some changes. We have tried our best to improve and made some changes in the manuscript.Our response and modifications to the questions raised are as follows: 1. We have added descriptions and images of the evtol at (80-83,p2). 2.We apologize for poor quality of figure 2, but Figure 2 is only provided to reflect the error between the wind tunnel data and the mathematical model. And these data are all from reference bellow, and we may not be able to obtain more ideal data. Harendra P B, Joglekar M J, Gaffey T M, et al. V/STOL tilt rotor study. Volume 5: A mathematical model for real time flight simulation of the Bell model 301 tilt rotor research aircraft[R]. 1973. 3.The reason for taking the decimal point to 4 places in the table is that the relative error and coefficient of determination of some networks are relatively similar between 2 and 3 places, and higher accuracy can reflect their differences. 4. We concur with the view that the addition of a parameter list may prove beneficial. However, we have already provided a detailed explanation of the meaning of the corresponding parameters in the aforementioned article. Given the constraints of length, we have chosen not to include a parameter list.But we have confirmed all possible parameters that may not have been indicated and completed the modifications(91-94,p3).We hope that this decision will be understood. 5. Thank you very much for your recognition of our prediction results, but if we use one panel for each graph. It may lead to a lengthy article, so we have only added a more detailed description of the charts at (285-288,p9). 6.We may have missed a description of the current situation of ANN in the introduction, we have adopted the reference you recommended and now added some improvements. (50-53,p2) 7.We appreciate the reviewer’s insightful suggestion and agree that it would be useful to provide specific information about WT tests. However, such an analysis is beyond the scope of our paper, which aims only to show that the application of neural network methods in the aerodynamic modeling of evtol.But it will be reflected in our future research. 8.Our ANN architecture is mainly based on Python, but we have added corresponding descriptions at (179-182,p6). Best regards

Reviewer 3 Report
Comments and Suggestions for Authors
The study proposes to address the aerodynamic characteristics of electric vertical take-off and landing (evtil) aircraft using a neural network model that was created based on wind tunnel testing. The results of the simulations showed a relatively high accuracy of the predicted aerodynamic characteristics, which can be appropriately used when designing electric aircraft with vertical take-off. The level of study would enhance practical validation of theoretical results obtained from a neural network model with practical flight tests of an electric vertical take-off aircraft, especially in transient and unstable flight modes.
The work is processed at a good level. Graphic outputs are clear, well processed. Minor shortcomings are in the less transparent description of the parameters used in graphs and equations. For the study, the authors used 23 literary sources, 65% of whom are under 5 years of age.
Considering the prospective use of the proposed method of modeling the properties of flying devices, I recommend publishing the article after eliminating the formal shortcomings and after conducting the recommended tests.
Author Response
Dear reviewer: Thank you for your decision and constructive comments on my manuscript. We have carefully considered the suggestion of Reviewer and make some changes. We have tried our best to improve and made some changes in the manuscript.Our response and modifications to the questions raised are as follows: 1.We concur with the view that the addition of a parameter list may prove beneficial. However, we have already provided a detailed explanation of the meaning of the corresponding parameters in the aforementioned article. Given the constraints of length, we have chosen not to include a parameter list.But we have confirmed all possible parameters that may not have been indicated and completed the modifications(91-94,p3).We hope that this decision will be understood. 2.We appreciate the insightful suggestions from the reviewers and agree that providing specific information about actual flight testing is useful. However, such an analysis is beyond the scope of our paper, which aims only to show that the application of neural network methods in the aerodynamic modeling of evtol.But it will be reflected in our future research. Best regards

Round 2
Reviewer 1 Report
Comments and Suggestions for Authors
Dear Authors
You have responded to the most of my suggestions.
Best regards
Reviewer 2 Report
Comments and Suggestions for Authors
The manuscript has been improved so I am accepting the manuscript in the present form.